# Applying behavioural economics principles to increase demand for free HIV testing services at private doctor-led clinics in Johannesburg, South Africa: A randomised controlled trial

**Simamkele Bokolo**[1]*, **Suzanne Mabaso**[2], **Wentzel Kruger**[2], **Preethi Mistri**[1], **Laura Schmucker**[3], **Candice Chetty-Makkan**[1], **Sophie J. S. Pascoe**[1], **Alison Buttenheim**[4], **Harsha Thirumurthy**[3], **Lawrence Long**[1,5]

**1** Faculty of Health Sciences, Health Economics and Epidemiology Research Office, University of the Witwatersrand, Johannesburg, South Africa, **2** Foundation for Professional Development (FPD), Pretoria, South Africa, **3** Department of Medical Ethics and Health Policy, Perelman School of Medicine, University of Pennsylvania, Philadelphia, Pennsylvania, United States of America, **4** Department of Family and Community Health, School of Nursing, University of Pennsylvania, Philadelphia, Pennsylvania, United States of America, **5** Department of Global Health, Boston University School of Public Health, Boston, Massachusetts, United States of America

* sbokolo@heroza.org

**Data Availability Statement:** Data underlying the findings is available on the Open Science

## Abstract

Expanding free HIV testing service (HTS) access to include private clinics could increase testing rates. A donor funded programme, GP Care Cell, offered free HIV testing at selected private doctor-led clinics but uptake was low. We investigated whether HTS demand creation materials that used behavioural economics principles could increase demand for HIV testing at these clinics. We conducted a randomised controlled trial in Johannesburg, South Africa (January-April 2022) distributing brochures promoting HTS to adults in five private doctor-led clinic catchment areas. Individuals were randomised to receive three brochure types: (1) "Standard of care" (SOC) advertising a free HIV test and ART; (2) "Healthy lifestyle screening" promoted free low-cost health screenings in addition to HTS; and (3) "Recipient of care voucher" leveraged loss aversion and the endowment effect by highlighting the monetary value of free HTS. The primary outcome was presenting at the clinic following exposure to the brochures. Logistic regression compared outcomes between arms. We found that of the 12,129 brochures distributed, 658 were excluded because of errors or duplicates and 11,471 were analysed. About 59% of brochure recipients were male and 50,3% were aged 25–34 years. In total, 448 (3.9%) brochure recipients presented at the private doctor-led clinics of which 50.7% were males. There were no significant differences in clinic presentation between the healthy lifestyle screening and SOC arm (Adjusted Odds Ratio [AOR] 1.02; 95% CI 0.79–1.32), and similarly between the recipient of care voucher and SOC arm (AOR 1.08; 95% CI 0.84–1.39). Individuals were more likely to attend centrally-located clinics that had visible HTS branding (AOR = 5.30; 95% CI 4.14–6.79). Brochures that used behavioural insights did not increase demand for HTS at private doctor-led clinics. However, consistent distribution of the brochures may have potential to increase HIV testing uptake at highly visible private doctor-led clinics.

Framework data repository on https://osf.io/uqn9c/
.

**Funding:** The GPCC was co-funded by the Gauteng Department of Health and through a sub-award under Anova Health Institute from their US President's Emergency Plan for AIDS Relief (PEPFAR) / United States Agency for International Development (USAID) APACE grant (Agreement Number 72067418CA00023). The study overlaid the existing GPCC investment through support from the University of Pennsylvania via a Bill and Melinda Gates Foundation grant (INV008318). Long was partially supported by the National Institute of Mental Health of the National Institutes of Health under grant number K01MH119923. The content is solely the responsibility of the authors and does not necessarily represent the official views of the funders or the United States Government.

**Competing interests:** The authors have declared that no competing interests exist.

## Backgrounds

HIV testing is the gateway to HIV prevention and treatment programmes and is essential for achieving epidemic control as reflected in the global UNAIDS 95-95-95 goals. These goals are aimed at ending the AIDS pandemic by ensuring that 95% of all people living with HIV know their HIV status, 95% of all people diagnosed with HIV infection receive antiretroviral therapy (ART), and 95% of all people receiving ART are virally suppressed [1]. Globally, 84% of people living with HIV (PLHIV) know their HIV status [2] while in South Africa over 92% of PLHIV know their status, putting South Africa on track towards reaching its first 95 target by 2030. However, the overall success of the HIV testing programs hides disparities in testing coverage; rates remain lower in certain populations at increased vulnerability to HIV infection including gay men and other men who have sex with men, female sex workers and people who inject drugs [3]. As such, there is a need to consider innovative approaches to promoting HIV testing in South Africa that incorporate diverse stakeholders in the field to access the populations as yet unreached by the traditional public clinic service model.

Current testing approaches, which are largely focussed on increasing testing accessibility within public health facilities and mobile testing points supported by non-governmental organisations, may exclude certain populations at increased risk of HIV [4,5]. Expanding testing delivery models to include private doctor-led clinics may be preferred by certain clients because of perceived increased trust and quality in the services of a private provider, which might result in clients being more likely to follow through with testing and treatment [6,7]. A range of studies present evidence on the potential role of leveraging the private health sector in enhancing the HIV response [8]. In South Africa the National Health Insurance (NHI) programme will leverage private clinics to provide primary health care, that includes HIV testing and treatment, making this an important model to evaluate [9]. The NHI Bill was signed into law on 15 May 2024 and aims to provide universal health coverage to all South Africans through a single pool of funding for private and public healthcare providers [10]. However, there are still limited studies that evaluate this type of model in low- and middle-income countries (LMICs) and even fewer studies evaluating demand creation for these services [11,12].

Solely increasing access to testing models may be insufficient since HIV-related health behaviours are often subject to various biases in individual decision-making [13,14]. Present bias is among the barriers to health behaviour, as many individuals place a strong emphasis on immediate costs and benefits when making decisions [15,16]. In healthy individuals, the benefits of engaging in HIV prevention and treatment are not immediate and individuals tend to postpone accessing these services. In a study conducted in Uganda with PLHIV on ART, 36% were classified as present-biased as they preferred an immediate hypothetical reward over a large reward that would be available a year later [17]. In the same study, the present-biased participants were also less likely to adhere to ART. In addition to present bias, the framing of behavioural messages or interventions determines the acceptability and uptake of the intervention. Another study in rural Uganda that assessed the effectiveness of incentive strategies to promote HIV testing among men used various behaviourally informed framing styles such as loss framing in which individuals were offered a reward with the condition that they perform the desired behaviour [18]. Similar studies have been conducted in Uganda, Tanzania and other countries using behaviourally informed interventions such as small incentives and planning prompts to improve HIV testing and have shown the potential of low-cost interventions to increase the uptake of HIV testing services [19–22]. In the context of this study, this means that in addition to having modalities that offer services to hard to reach populations it is also important to help potential clients make the decision to test before they get to the site.

Behavioural economics suggests several strategies to address these biases and simplify the individuals' decision making.

We leveraged an existing private doctor-led clinic network (GP Care Cell (GPCC) programme) offering free HIV testing and treatment in South Africa to test whether behaviourally informed demand creation material could increase HIV testing uptake. The demand creation material used the behavioural economics principles of loss aversion, the endowment effect and framing to improve HTS uptake decision making.

## Methods

### Study setting and design

The GPCC programme in South Africa was set up in the Johannesburg Health District and supported by the Foundation for Professional Development and PPO Serve in partnership with the Gauteng Department of Health to expand access to HIV testing services and antiretroviral therapy (ART) for uninsured PLHIV at private doctors outside of public sector health facilities [23]. The GPCC was a NHI demonstration project designed to provide clients who were not insured with the option of testing for HIV, and if positive, initiating and receiving HIV treatment at a private doctor-led clinic. The services provided were at no cost to the clients and so testing in these clinics meant that there was a direct pathway for treatment for those found to be living with HIV. However, early unpublished findings from this programme showed poor HIV testing demand at the clinics, partly due to limited investments by the clinics in community demand creation.

This randomised controlled trial (RCT) took place in Johannesburg, South Africa. Johannesburg is South Africa's most populous city, with 5.5 million residents, most (84%) of which are uninsured for healthcare services [24]. In 2020, the HIV prevalence was 13% in Johannesburg with the highest incidence recorded among females compared to males of all age groups [25]. The study was conducted in the catchment areas of five private doctor-led clinics that were part of the GPCC programme. The selected private doctor-led clinics had large catchment areas yet faced low demand for HIV testing. All the clinics were located at high transit settings with large numbers of pedestrians (two in shopping centres, one at a train station, and two in the Central Business District) and reported high proportions of cash-paying clients (40–60% of patient load). Two of the clinics had branded gazebos erected in the clinic for the purposes of HIV testing, whereas others only offered clinic-based testing (in which testing was done inside the clinic building). Branded gazebos increased the clinic's visibility. Operating hours of these clinics ranged between 8:00 and 17:00 on weekdays and were also open on Saturdays.

### Intervention design

This study included three types of demand creation brochures that were given to individuals in the catchment areas: 1) "standard of care" brochure 2) "healthy lifestyle screening" brochure and 3) "recipient of care voucher" brochure (S1 Fig). The standard of care brochure advertised a free HIV test and antiretroviral therapy (ART). It also provided details about the GPCC programme and a summary of all services provided. The healthy lifestyle screening brochure sought to address stigma associated with HIV testing by bundling the free HIV test with other less stigmatised health screenings such as blood pressure, glucose and body mass index. The healthy lifestyle screening brochure was also framed to simplify participants' decision making as it may be easier for them to accept an HIV test as part of a comprehensive screening offer. The recipient of care voucher brochure advertised HTS by designing the brochure to resemble a voucher that entitled the holder to an HIV test at the participating private doctor-led clinic

which was worth approximately ZAR 100 (US$7) if they presented before the expiration date (~2 weeks). The recipient of care voucher brochure leveraged BE principles of loss aversion (individuals tend to want to avoid losses more strongly than they enjoy equivalent gains) and the endowment effect (people tend to value things that they own more than something that does not yet belong to them) [26]. Our hypothesis was that individuals who receive the voucher brochure will try to avoid losing its value by redeeming it for an HIV test before it expires. HIV testing is typically provided at no cost at public health clinics in South Africa but is typically not free in the private sector. Attaching a monetary value to the HIV test increases the perceived value of a service that many South Africans may consider low value because it is offered free of charge. All the study brochures included a WhatsApp number that could be used to book an appointment. This was mainly for those who wished to make an appointment or find other clinics they could go to, however, all brochures included the text "Walk-ins welcome" to accommodate participants who might not have had access to WhatsApp or a phone.

Prior to data collection, prototyping of the three brochures took place at one participating private doctor-led clinic with eleven participants that were in the waiting room. The aim of prototyping was to receive feedback on the format, structure, language and understanding of the brochure content. Overall, the brochures received positive ratings with the primary concerns relating to wording, understanding the message and understanding the language. Feedback from the prototyping informed revisions to the brochures prior to study implementation.

### Randomisation and data collection

The brochures were distributed to adults aged ≥18 years in the immediate catchment area of each clinic. The brochures were available in three languages (English, IsiZulu, and SeSotho) that are common in Johannesburg. Brochures were randomly allocated in a 1:1:1 ratio to each participant using a randomisation framework to determine the order in which brochures were distributed. In order to ensure that the randomisation was balanced across the different language options, we used small block sizes (15) to ensure that there was equal assignment across the study arms (brochures) and uniform distribution of the 3 brochures in each of 3 languages. The brochures were placed in sealed envelopes (the fieldworkers were blinded to the brochures) which were labelled with the randomisation code and distributed in order. During distribution, the individually randomised brochures were ordered into three language packs and fieldworkers distributed the brochures to participants in the order of the language packs depending on the participants' language preference. Fieldworkers used a recruitment script to engage with potentially eligible individuals and, based on willingness and language preference, handed them the next envelope with a brochure. The fieldworker and a reminder on the envelope prompted individuals to return with the brochure to the clinic to access services. The fieldworker recorded basic demographics (age group, gender), information on the demand creation setting, and the randomisation number(s) of distributed envelopes on the REDCap system or on a paper-based version.

All individuals presenting for HIV testing services at the study sites (that is the selected GPCC sites) during the study period notified the clinical team on how they heard about the service and handed in any brochures that they had received. Individuals demonstrating exposure (presenting a study brochure) or self-reporting exposure (reporting having received a brochure from a fieldworker or other contact but not having the brochure present) were invited to participate in the study. Individuals self-reporting exposure were asked to describe the brochure they were exposed to and shown the described brochure to confirm exposure. The primary outcome for this study was presentation at the private doctor-led clinic and participants consenting to be part of the study were included even if they later refused to receive HTS or

other services. The secondary outcomes included HIV testing uptake, HIV-positivity among participants who get tested for HIV, and linkage to care among newly diagnosed individuals.

We anticipated that about 5% of participants in the standard of care group will present at the private doctor-led clinic. This was a conservative assumption based on the experiences of similar campaigns in which individuals are approached in the community. With 2,400 adults randomised to each of the three study arms (total sample size of 7,200 adults), we would have >80% power (alpha = 0.05, 2- sided) to detect a difference of at least 2 percentage points in attendance at private doctor-led clinics. However, since there was a possibility that a larger proportion of participants in the standard of care group would present at the private doctor-led clinics, we selected a sample size of 4,000 adults per arm (12,000 total) as this ensured that there is 80% power to detect a difference of at least 2 percentage points. A small difference in the primary outcome would be meaningful given the low cost of the demand creation interventions.

## Data analysis

During the data cleaning process, we identified a number of data entry errors and duplicate entries in the dataset. For the primary analysis we addressed duplicates by: a) including only the first entry of any observation that appeared to be a true duplicate of the same participant (i.e. duplicate randomisation code with the same demographics), and b) including both entries if it appeared that they were different participants (i.e. duplicated randomisation codes yet the demographic information was different). All other data entry errors that could not be resolved were excluded from the final analytic dataset. We tested to see whether this decision influenced the results by running an analysis where all duplicates were excluded.

The primary outcome was presentation at the private doctor-led clinic after receipt of the brochure within the 4-month period of data collection. The distribution of brochures at all clinics ended at least 2 weeks before the end date of the data collection period at the private doctor-led clinics ended. This method allowed time for participants who had received brochures to present at the clinic. The period for presenting at the clinic for HTS was contingent upon the receipt of the brochure and ranged from a minimum of 2 weeks to a maximum of 4 months. We used chi-square tests for categorical variables such as age group, gender, clinic and language to determine frequencies and associated proportions. Multivariable logistic regression models were used to determine factors independently associated with presenting at a private doctor-led clinic. In the analysis, we adjusted for gender, age group, clinic and language fixed effects to control for possible confounders. We present data as odds ratio (OR) and adjusted odds ratios (AOR) with 95% confidence intervals (CI). We analysed data using STATA version 17 (StataCorp, Texas, USA).

The sensitivity analysis (excluding duplicates) is reported in the supplementary material, refer to S1 Table. For the supplementary analysis, we compared clinic presentations across clinic locations. We also created a variable to compare clinic presentation between clinics with high visibility (1 and 5) versus those with limited visibility (2, 3 and 4). We used the CONSORT checklist when writing our report (see S1 Checklist).

## Ethics approval and consent to participate

The study protocol (see S1 Protocol) was approved by the University of Witwatersrand, University of Pennsylvania and Boston University and provided a waiver of informed consent during the distribution of brochures. Handing out brochures is a standard demand creation strategy and no personal identifiers were collected. The study could not have been conducted if consent were obtained. However, participants presenting at the private doctor-led clinic for

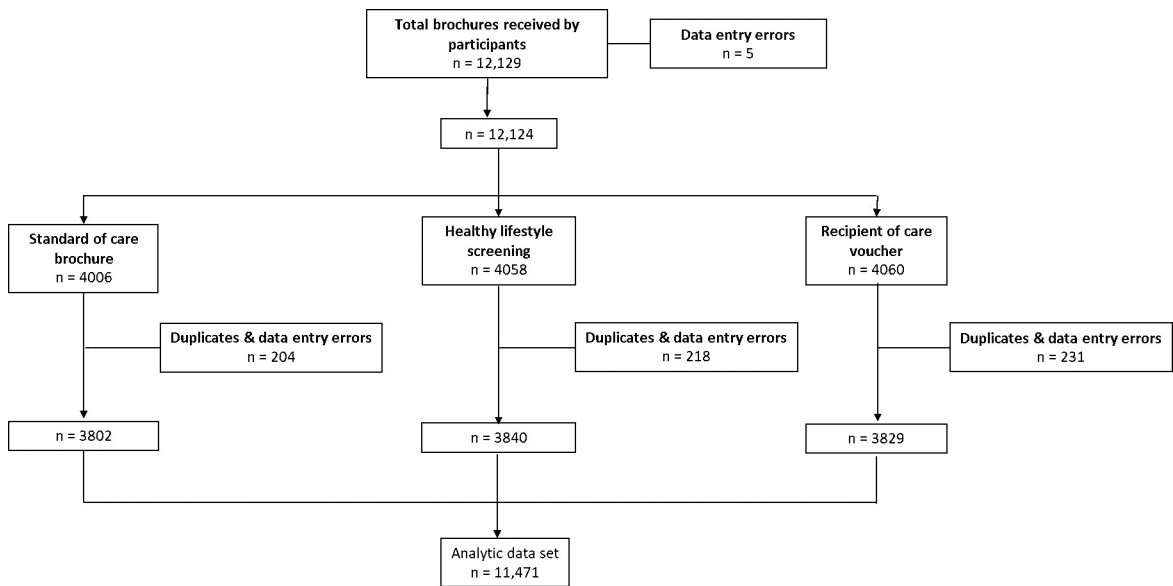

**Fig 1. CONSORT diagram describing the total number of brochures received by participants across the three study arms (n = 11,471).**

services were requested to provide written consent before any data were collected and signed the HTS consent form before any medical procedures were done (standard of care). The trial was registered with the South African National Clinical Trials Registry (https://sanctr.samrc.ac.za/), registration number: DOH-27-032022-9053.

## Results

A total of 12,129 brochures were distributed between January and April 2022. We excluded 658 entries as these were duplicates and data entry errors (Fig 1). The distribution of duplicates and data errors were not different across the study arms. The final dataset included 11,471 of the distributed brochures.

Of the 11,471 brochures, more than half (59.1%) were distributed to males and the majority (66.8%) to young adults (≤35 years old). The number of brochures distributed across the 5 participating private doctor-led clinics were similar and more English brochures (71.2%) were distributed compared to the other languages (See Table 1).

Fig 2 shows the proportion of individuals presenting at the private doctor-led clinic by study arm. In regression analyses (Table 2), there were no significant differences in clinic presentation between the healthy lifestyle screening and SOC arm (Adjusted Odds Ratio [AOR] 1.02; 95% CI 0.79–1.33), and similarly between the recipient of care voucher and SOC arm (AOR 1.08; 95% CI 0.84–1.40). Results were robust to the exclusion of all duplicates (see S1 Table). When the clinics were grouped based on site characteristics, individuals were more likely to attend private doctor-led clinics that had high visibility for HTS (AOR = 5.30; 95% CI: 4.14–6.79) across all study arms (see S3 Table). This included private doctor-led clinics that had gazebos and banners advertising the clinic and services provided set-up in visible areas of the clinic. This association did not vary by study arm.

## Discussion

HIV testing uptake at private doctor-led clinics in South Africa remains low despite free provision of testing services. Our study found that distributing brochures that used behavioural

**Table 1. Characteristics of participants who received a brochure to attend HTS at a private clinic between 24 January 2022 to 15 April 2022.**

| | | Number of participants receiving brochures | | | |
|---|---|---|---|---|---|
| | | Total brochures received by participants (n = 11 471) | Standard of care brochure (n = 3802) | Healthy lifestyle brochure (n = 3840) | Recipient of care voucher brochure (n = 3829) |
| | | n (%) | n (%) | n (%) | n (%) |
| **Language distribution** | English | 8165 (71.2) | 2716 (71.4) | 2720 (70.8) | 2729 (71.3) |
| | IsiZulu | 2656 (23.2) | 874 (23.0) | 899 (23.4) | 883 (23.1) |
| | SeSotho | 650 (5.7) | 212 (5.6) | 221 (5.8) | 217 (5.7) |
| **GP practice catchment** | Clinic_1 | 2205 (19.2) | 723 (19.0) | 748 (19.5) | 734 (19.2) |
| | Clinic_2 | 2354 (20.5) | 787 (20.7) | 788 (20.5) | 779 (20.3) |
| | Clinic_3 | 2320 (20.2) | 766 (20.2) | 779 (20.3) | 775 (20.2) |
| | Clinic_4 | 2211 (19.3) | 731 (19.2) | 732 (19.1) | 748 (19.5) |
| | Clinic_5 | 2381 (20.8) | 795 (20.9) | 793 (20.7) | 793 (20.7) |
| **Gender** | Male | 6774 (59.1) | 2246 (59.1) | 2293 (59.7) | 2235 (58.3) |
| | Female | 4601 (40.1) | 1532 (40.3) | 1512 (39.4) | 1557 (40.7) |
| | Other | 24 (0.2) | 8 (0.2) | 6 (0.1) | 10 (0.3) |
| | Missing | 72 (0.6) | 16 (0.4) | 29 (0.8) | 27 (0.7) |
| **Age group** | 18–24 years | 1890 (16.5) | 654 (17.2) | 612 (15.9) | 624 (16.3) |
| | 25–34 years | 5774 (50.3) | 1901 (50.0) | 1911 (49.8) | 1962 (51.2) |
| | 35–44 years | 3066 (26.7) | 1015 (26.7) | 1060 (27.6) | 991 (25.9) |
| | ≥ 45 years | 669 (5.8) | 216 (5.7) | 228 (5.9) | 225 (5.9) |
| | Missing | 72 (0.6) | 16 (0.4) | 29 (0.8) | 27 (0.7) |

Among participants receiving brochures, 448 (3.9%) individuals presented at a private doctor-led clinic for services within 4 months and 50.7% of those presenting were males (see S2 Table). Most participants presented at clinic 1 (54.5%) and clinic 5 (35.7%) which were centrally located and had high visibility (centrally located with visible branding) compared to other clinics.

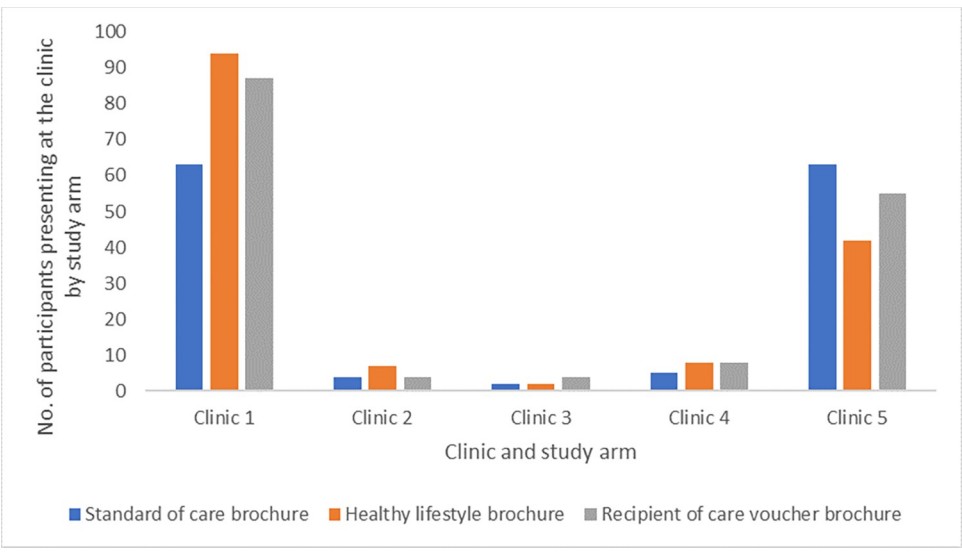

**Fig 2. Proportion of individuals presenting at the private doctor-led clinic by study arm.**

**Table 2. Effect of intervention on odds of presenting at the GP practice with study arm.**

| Study arms | No. of participants receiving the brochures | No. of participants presenting at the clinic | % (95% CI) | Unadjusted OR (95% CI) | Adjusted OR* (95% CI) |
|---|---|---|---|---|---|
| Standard of care | 3802 | 137 | 3.6% (3.03–4.25%) | 1 [reference] | 1 [reference] |
| Healthy lifestyle screening | 3840 | 153 | 4.0% (3.39–4.65%) | 1.11 (0.88–1.40) | 1.02 (0.79–1.33) |
| Recipient of care voucher | 3829 | 158 | 4.1% (3.52–4.81%) | 1.15 (0.91–1.45) | 1.08 (0.84–1.40) |

*Adjusted for gender, age group, clinic and language.

economics principles to promote the free HIV testing services in the clinic catchment areas did not increase HIV testing uptake compared to a traditional brochure. However, we found that distributing brochures in these locations outside of traditional health care facilities reached more men than women across all clinic locations. Supplementary findings showed that participants were more likely to present at private doctor-led clinics that were centrally located with high visibility. These findings highlight the various factors influencing the uptake of HIV services at private doctor-led clinics. Establishing the effectiveness of the specific behavioural insights used to inform the intervention brochures is important in understanding the overall potential of behavioural economics in improving HIV outcomes.

There is limited research that uses randomised trials to rigorously evaluate alternative demand creation strategies to reach people for HIV testing in LMICs. Behaviourally informed brochures have been evaluated in other settings including studies promoting physical well-being [27] with minimal focus on using BE principles to design brochures for HIV testing. In our study, we used a novel non-incentive intervention (recipient of care voucher) while other studies in rural Uganda and South Africa focused on using financial incentives to reach people for HIV testing [28,29]. These studies reported increased HIV testing uptake among participants in the incentives arm and were successful in reaching certain population groups. We further tested the healthy lifestyle screening which bundled HIV testing with other non-communicable disease screening to address the stigma associated with HIV testing. This approach was used in the public sector and adopted in other countries [30–32], although experimental evaluations of such approaches remain limited. Even though our results did not show an increase in HIV testing when compared to standard of care, other studies support the use of behavioural insights to inform interventions promoting HTS uptake [33]. While our interventions did not increase demand for HTS uptake among the general population, the intervention may be useful in expanding access to populations that are hard to reach through traditional testing models.

The distribution of men (50% men to 50% women) presenting for testing is still higher than is typically reported through traditional HIV testing campaigns [34–36], 45% men vs 59% women aged 15–49 years were reported to have been tested in 2016 [37]. Several factors could have influenced the higher uptake of HTS among males. The novel approach to promoting HIV testing in private, doctor-led clinics may have been particularly appealing to men. The distribution sites were also areas where men were more likely to pass through as they go or return from work. However, we also acknowledge that the results may be influenced by the manner in which the brochures were distributed and who was more willing to accept the brochures. The low participation of men in HIV testing programmes generally has been attributed

to the fear of damaging their social reputation, potential of community rejection and emotional distress likely to be caused by a HIV-positive result [38]. The private doctor-led clinic testing model, though not directly linked to our interventions, addresses the disparities in HIV testing and holds potential to reach sub-populations that may not be reached through public sector testing modalities.

Supplementary findings showed that participants were 5.3 times more likely to present for services at clinics that were centrally located with high visibility. Studies have established that increased accessibility and affordability of health services is associated with higher clinic attendance [39–41]. Reducing hassle factors to accessing HIV testing services has potential to improve clinic attendance and it is possible that had the location characteristics of all the private doctor-led clinics matched the clinics with high presentation rates (centrally located with high visibility), there would be increased access across all. It is also possible that considering the private doctor-led clinic location may also attract more clients thereby increasing HIV testing demand.

The study had some key strengths and limitations worth highlighting. This study was embedded within the existing GP Care Cell program which reduced the study-related costs of setting up an independent project and facilitated the process of co-designing the intervention with the implementing partner (FPD) applying behavioural economics principles. However, we noted that the standard demand creation distribution before the study was not the same as the study distribution which might partly explain the results presented. Since the first step of data collection (brochure distribution) took place at the private doctor-led clinics catchment areas, there were safety concerns that made it difficult for some data teams to record data on REDCap in real time and had to opt for paper-based data collection; this approach led to some of the data entry errors we reported in the paper. Despite these limitations we reached our set target and data collection for this RCT was completed in 4 months showing that it is possible to pair strong evaluation design with rapid results.

## Conclusion

Expanding access to HIV testing is central in reaching the UNAIDS 95-95-95 case finding targets. However, the success of the testing target in the UNAIDS 95-95-95 strategy masks the disparities in HIV testing coverage across populations. Services in the public health sector are unlikely to accommodate all populations and as we expand access to HIV testing services it is important to consider the role of the private sector in accelerating service delivery to bridge the gap. This requires exploring options to integrate HIV testing provision into the private health sector in line with the envisioned NHI policy and generating sufficient demand to attract the targeted populations.

## Supporting information

**S1 Checklist.**
(DOCX)

**S1 Fig. HTS demand creation material\*.**
(DOCX)

**S1 Table. Sensitivity analysis of Odds ratios (95% CI) from exploratory logistic regression results comparing SOC, healthy lifestyle brochure and care recipient voucher brochure arms\*.**
(DOCX)

**S2 Table. Proportion of individuals presenting at the GP practice over the total number of brochures distributed by study arm.**
(DOCX)

**S3 Table. Effect of intervention on odds of presenting at the GP practice with sociodemographics\*.**
(DOCX)

**S1 Protocol.**
(PDF)

## Acknowledgments

We would like to thank the Foundation for Professional Development for collaborating in the study and for the operational support for this study. We would also like to thank the participating General Practitioners who were part of the GP Care Cell programme for supporting this study.

## Author Contributions

**Conceptualization:** Simamkele Bokolo, Suzanne Mabaso, Preethi Mistri, Laura Schmucker, Candice Chetty-Makkan, Sophie J. S. Pascoe, Lawrence Long.

**Data curation:** Suzanne Mabaso, Wentzel Kruger, Candice Chetty-Makkan.

**Formal analysis:** Simamkele Bokolo, Preethi Mistri, Laura Schmucker, Candice Chetty-Makkan, Sophie J. S. Pascoe, Alison Buttenheim, Harsha Thirumurthy, Lawrence Long.

**Funding acquisition:** Sophie J. S. Pascoe, Harsha Thirumurthy.

**Investigation:** Suzanne Mabaso, Candice Chetty-Makkan, Sophie J. S. Pascoe, Harsha Thirumurthy, Lawrence Long.

**Methodology:** Simamkele Bokolo, Suzanne Mabaso, Laura Schmucker, Candice Chetty-Makkan, Sophie J. S. Pascoe, Harsha Thirumurthy, Lawrence Long.

**Project administration:** Simamkele Bokolo, Suzanne Mabaso, Preethi Mistri, Laura Schmucker, Candice Chetty-Makkan.

**Resources:** Suzanne Mabaso.

**Supervision:** Candice Chetty-Makkan, Sophie J. S. Pascoe, Alison Buttenheim, Harsha Thirumurthy, Lawrence Long.

**Validation:** Simamkele Bokolo, Wentzel Kruger, Preethi Mistri, Candice Chetty-Makkan, Lawrence Long.

**Visualization:** Simamkele Bokolo, Laura Schmucker, Candice Chetty-Makkan, Sophie J. S. Pascoe, Alison Buttenheim, Harsha Thirumurthy, Lawrence Long.

**Writing – original draft:** Simamkele Bokolo.

**Writing – review & editing:** Suzanne Mabaso, Preethi Mistri, Laura Schmucker, Candice Chetty-Makkan, Sophie J. S. Pascoe, Alison Buttenheim, Harsha Thirumurthy, Lawrence Long.

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
