## [Decision Letter · Decision Letter 0]

7 May 2024

PGPH-D-23-01678

Applying behavioural economics principles to increase demand for free HIV testing services at private doctor-led clinics in Johannesburg, South Africa: A randomised controlled trial

Dear Dr. Bokolo,

Thank you for submitting your manuscript to PLOS Global Public Health. After careful consideration, we feel that it has merit but does not fully meet PLOS Global Public Health’s publication criteria as it currently stands. Therefore, we invite you to submit a revised version of the manuscript that addresses the points raised during the review process.

We look forward to receiving your revised manuscript.

Kind regards,

Carmen S. Christian, PhD

Academic Editor

Journal Requirements:

2. Please ensure you have included the registration number for the clinical trial referenced in the manuscript.

3. Please provide separate figure files in .tif or .eps format only and remove any figures embedded in your manuscript file. Please also ensure all files are under our size limit of 10MB.

4.  We have noticed that you have uploaded Supporting Information files, but you have not included a list of legends. Please add a full list of legends for your Supporting Information files after the references list.

5. In the online submission form, you indicated that "Data is available upon reasonable request". All PLOS journals now require all data underlying the findings described in their manuscript to be freely available to other researchers, either 1. In a public repository, 2. Within the manuscript itself, or 3. Uploaded as supplementary information.

Additional Editor Comments (if provided):

Reviewers' comments:

Reviewer's Responses to Questions

**Comments to the Author**

1. Does this manuscript meet PLOS Global Public Health’s publication criteria? Is the manuscript technically sound, and do the data support the conclusions? The manuscript must describe methodologically and ethically rigorous research with conclusions that are appropriately drawn based on the data presented.

Reviewer #1: Yes

Reviewer #2: Yes

2. Has the statistical analysis been performed appropriately and rigorously?

Reviewer #1: Yes

Reviewer #2: Yes

3. Have the authors made all data underlying the findings in their manuscript fully available (please refer to the Data Availability Statement at the start of the manuscript PDF file)?

Reviewer #1: No

Reviewer #2: Yes

4. Is the manuscript presented in an intelligible fashion and written in standard English?

Reviewer #1: Yes

Reviewer #2: Yes

5. Review Comments to the Author

Reviewer #1: Thank you to the authors for the opportunity to review this article. It is clearly a substantial research project, and just needs some refinement on the writing and conveying of key concepts. I have made some suggestions below. There are also some larger methodological question regarding the piloting work.

Background

56 HIV testing is the gateway to HIV prevention and treatment programmes and is essential for achieving

57 epidemic control as reflected in the global UNAIDS 95-95-95 goals. Globally, 84% of people living with

58 HIV (PLHIV) know their HIV status1 while in South Africa over 92% of PLHIV know their status, putting

59 South Africa on track towards reaching its first 95 target by 2030.

Comment: you will need to explain to the reader what the 95-95-95 target is (including a citation), as anybody not familiar with the field will struggle to understand.

However, HIV testing rates remain low

60 in certain populations at risk of HIV infection including gay men and other men who have sex with men,

61 female sex workers and people who inject drugs2.

Comment: What is considered to be low? Also, are these groups specifically vulnerable?

In South Africa the proposed National Health

72 Insurance (NHI) programme will leverage private clinics to provide primary health care, which would

73 include HIV testing and treatment, making this an important model to evaluate8

Comment: Given that this is for an international audience, you will have to provide more elaboration here.

However, there are still

74 limited studies that evaluate this type of model in low- and middle-income countries (LMICs) and even

75 fewer studies evaluating demand creation for these services.

Comment: Do you have a reference for this?

Present bias is

79 among the barriers to health behaviour, as many individuals place a strong emphasis on

80 immediate costs and benefits when making decisions10,11.

Comment: you’ll need to elaborate on “present bias” for the reader. Also, perhaps elaborate on the studies you cite here. What do they say about present bias being a barrier to testing?

Overall comment of introduction/background: This is a behavioural study, but there is very little discussion of the behavioural principles that you are studying and the evidence on these principles or how they tend to affect health outcomes (i.e. framing effects, present bias). Please build on this.

Methods:

This randomised controlled trial (RCT) took place in Johannesburg, South Africa

Comment: Please elaborate on the overall and health sector context of Johannesburg for international readers.

113 Intervention design

Comment: It’s not exactly clear to me why recipients of the pamphlets would need to bring the pamphlet with to their clinic visits, other than with the “voucher model”. How do you know that people weren’t incentivised to go to the clinic and did not bring their pamphlet along? I’m sure this was controlled for, but it’s not coming out clearly in the write-up.

The recipient of care

126 voucher brochure leveraged BE principles of loss aversion (individuals tend to want to avoid

127 losses more strongly than they enjoy equivalent gains) and the endowment effect (people tend

128 to value things that they own more than something that does not yet belong to them)

Comment: How are these linked to the pamphlet (more detail is necessary), and how will they overcome the barriers.

we used

143 small block sizes (15) to ensure uniform distribution of the 3 brochures in each of 3 languages

Comment: Sorry, it’s unclear what this means. Was randomization done at an individual level?

218 Among participants receiving brochures, 448 (3.9%) individuals presented at a private doctor-led

219 clinic for services within 4 months and 50.7% of those presenting were males (see

220 supplementary table 1).

Comment: Is this your final sample size? Is this enough to have power? Also, what are the demographic differences between the group that used the pamphlets, and the ones that did not (e.g. the whole 11000 sample)? Also, please see earlier point about linking pamphlets to patients.

Methodology overall: Did you do any piloting work that showed that present bias was a problem in the context of your clinics? Or what the main barriers to access in this catchment area are?

Discussion

249 Establishing the effectiveness of the specific behavioural insights used to inform the intervention

250 brochures is important in understanding the overall potential of behavioural economics in

251 improving HIV outcomes.

Comment: This point does not flow from the previous point on the visibility of clinics, and it’s a bit out of place and unsubstantiated as it stands here.

270 Interestingly, 59% of the demand creation materials distributed were received by males and just

271 over 50% of those presenting at the private doctor-led clinic for services were also male.

Comment: This should be in the results section, not the discussion section. The rest of the paragraph is very interesting and should stay in the discussion section and refer to this finding in the results. I wonder if “interestingly” is an appropriate word to use in a journal context.

Reviewer #2: This is an important paper that advances the value of rapid behavioural insight tests to improve HIV programmes.

The comments are structured with an extract from the text, and a request for further clarity

“All the clinics were located at high transit settings with large numbers of pedestrians (two in shopping centres, one at a train station, and two in the Central Business District)… Most participants presented at clinic 1 (54.5%) and clinic 5 (35.7%) which were centrally located and had high visibility (centrally located with visible branding) compared to other clinics.”: There is opportunity to provide further information on site characteristics because they emerge as a key component of the results. For example, high visibility, and central location only appear in the results section, with little background into how these are defined

Individuals demonstrating exposure (presenting a study brochure) or self-reporting exposure (reporting having received a brochure from a fieldworker or other contact but not having the brochure present) were invited to participate in the study: Please clarify how study staff knew the arm to which participants were assigned to if they showed up without the brochure or envelope.

The primary outcome was presentation at the private doctor-led clinic after receipt of the brochure within the 4-month period of data collection - please clarify if this means that everyone had at least 4 months to make the clinic visit. If the “follow-up time” depended on when one received the brochure relative to the end date of data collection, please describe when the distribution of brochures stopped for each of the clinics. It is helpful to know this for putting the results in context (especially the secondary analysis where comparisons are made at the clinic level): if the sample size for Clinic X was reached earlier than Clinic Y, could that have given participants from Clinic X more opportunity to make the clinic visit?

However, we found that distributing brochures in these locations outside of traditional healthcare facilities reached more men than women across all clinic locations. It is not entirely clear how this finding is to be interpreted. Could this be an artefact of how study staff distributed the brochures since they had some control over who they approached to give the brochure?

Brochures: The brochures suggest that participants had to send a WhatsApp message to book an appointment. If so, where there any considerations for people without cell phones?

6. PLOS authors have the option to publish the peer review history of their article (what does this mean?). If published, this will include your full peer review and any attached files.

**Do you want your identity to be public for this peer review?** For information about this choice, including consent withdrawal, please see our Privacy Policy.

Reviewer #1: No

Reviewer #2: No

[NOTE: If reviewer comments were submitted as an attachment file, they will be attached to this email and accessible via the submission site. Ple

---

## [Editor Report · Decision Letter 1]

20 Jun 2024

Applying behavioural economics principles to increase demand for free HIV testing services at private doctor-led clinics in Johannesburg, South Africa: A randomised controlled trial

PGPH-D-23-01678R1

Dear Ms Bokolo,

We are pleased to inform you that your manuscript 'Applying behavioural economics principles to increase demand for free HIV testing services at private doctor-led clinics in Johannesburg, South Africa: A randomised controlled trial' has been provisionally accepted for publication in PLOS Global Public Health.

Best regards,

Carmen S. Christian, PhD

Academic Editor